# The Influence of Processing Conditions on the Quality of Bent Solid Wood from European Oak

Aleš Straže [1] , Jure Žigon [1] , Stjepan Pervan [2], Mislav Mikšik [2] and Silvana Prekrat [2,*]

1   Biotechnical Faculty, University of Ljubljana, Jamnikarjeva 101, 1000 Ljubljana, Slovenia; ales.straze@bf.uni-lj.si (A.S.); jure.zigon@bf.uni-lj.si (J.Ž.)
2   Faculty of Forestry, University of Zagreb, Svetošimunska cesta 23, 10000 Zagreb, Croatia; spervan@sumfak.unizg.hr (S.P.); mmiksik@sumfak.unizg.hr (M.M.)
*   Correspondence: sprekrat@sumfak.unizg.hr; Tel.: +385-1-235-2408

**Abstract:** Bending of solid wood from European oak is one of the most demanding technological processes due to its specific structural and physical properties and variability. We investigated the influence of wood moisture content (MC) and stiffness, determined by NDT, as well as previous drying methods on the bending ability of the wood. The best quality was obtained with bending specimens bent at a moisture content of at least 16% and quarter- or semi-quartersawn. The number of rejected specimens increased slightly when HF bending was used. Single-stage predrying of oak to a final MC of 8% resulted in a high rejection rate (>70%) regardless of drying technique. The acceptance rate was higher for less stiff specimens where the ratio of ultrasonic velocity in the straight ($v_S$) and bent region ($v_B$) was less than 0.5 ($v_B/v_S$).

**Keywords:** wood; solid wood bending; quality; nondestructive testing





## 1. Introduction

European oak (*Quercus robur* L., *Quercus petraea* (Matt.) Liebl.) is an important commercial tree species and a widely used industrial wood for a variety of products such as veneers, furniture, interior and exterior structures, and many other items [1]. In modern production of solid oak furniture, the need for complex spatial shapes of furniture elements is common. In these cases, the reduced mechanical properties of the wood in the transverse direction can be a limiting technological and applicable factor. Difficulties in the production of complex 3D solid wood shapes also arise from the low material yield during sawing and from the demanding technological processing [2,3]. One solution that overcomes these limitations is solid oak bending.

Solid wood bending has been practiced for centuries, and the quality of the bending is judged by the proper deformation achieved without apparent failure of the wood [4]. Of particular interest in solid wood bending are the effects of creep and, to a lesser extent, relaxation. Creep is greatly enhanced by the absolute value of moisture content (MC) and by the changes in MC of the wood under bending load due to the mechanosorptive effect [5–7]. The rate and extent of mechanosorptive deformation correspond to the extent of MC changes and are usually independent of the time during which the MC changes occurred [8,9].

In the practice of commercial wood bending, moistened solid wood is usually bent at higher temperatures, which leads to a corresponding plasticity of the material [10]. It is assumed that under such conditions the glass transition temperature ($T_g$), at which the modulus of elasticity (MOE) decreases significantly, is exceeded for some or all of the basic polymeric constituents of the wood. The highest $T_g$ in the dry state, above 200 °C, was found for cellulose, slightly lower, between 150 °C and 180 °C, for hemicelluloses, and below 150 °C for lignin [11,12]. However, several studies show that the transition from elastic to plastic mechanical behavior of wood occurs at temperatures well below 100 °C,

especially at higher moisture contents [13,14]. Most explanations for this phenomenon state that the low-molecular-weight water molecules in wood act as diluents and plasticizers [7].

However, the correct MC of specimens before bending is controversial. Conditioning the material in the range of 12 to 25% MC covers most bending applications and methods [15]. Specific bending radii and severity of deformation may require different MCs, achieved by different drying and presteaming methods [16,17]. Heat and moisture plasticizing by presteaming increases the compressibility of the wood by up to 40% of the compressive strain, but has virtually no effect (up to 2%) on the tensile ductility of the wood [10,15,18]. The softening treatment shifts the neutral axis of the bent parts towards the convex side, which is axially stressed during bending. This shift significantly improves the bending deformation of the wood. In addition, a metal band with end stop is used, which is wrapped around the convex side of the sample, a method discovered by Thonet in 1856 [18].

Due to the wide variability of chemical, structural, and mechanical properties of wood species, the bending quality of wood varies, so selection is important. The highest bending quality is obtained with straight-grained pieces, free of crossgrain, of generally fast-growing and less dense species [18,19]. Some studies also report that ring-porous woods generally give better results than diffuse-porous woods. In practice, quartersawn lumber is preferred for bending, but some sources also indicate that flatsawn lumber bends better in severe bends [10,16,18].

In this study, we aimed to investigate the possibility of bending of European oak wood, which is less commonly used in industrial practice because of the good bending properties of other European hardwoods. We were particularly interested in the influence of wood moisture content and previous drying methods on the bending ability of the wood. We also wanted to see if it was possible to determine the quality of the bending using non-destructive techniques.

## 2. Materials and Methods

### 2.1. Sampling

The European oak (*Quercus robur* L.) wood used in the study was obtained by sawing 1st-grade oak logs. The test pieces were 1300 mm long, 60 mm wide, and 38 mm thick and were straight-grained and sawn from heartwood. A total of 120 elements were taken from the mill's own lumberyard (Spin Valis d.d., Požega, Croatia; N 45.338396°, E 17.690455°, 311 m a.s.l.), with 40% of the elements oriented radially (R; quartersawn) and semi-radially (RT) and 20% of the elements oriented tangentially (T; flatsawn) (Figure 1). The initial moisture content of the samples was 56.3% (St.dev. = 10.2%; St.dev.—Standard deviation).

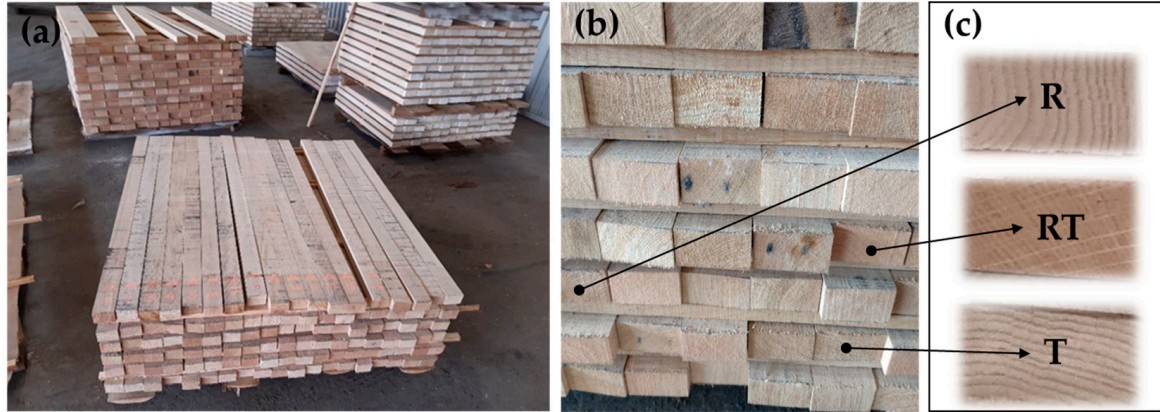

**Figure 1.** Sampling and sorting of green oak wood in the woodyard (**a**), wax protection of the front and back ends before drying (**b**), and orientation of the wood grain of test pieces (**c**).

*2.2. Material Processing*

2.2.1. Wood Drying

To optimize the organization of the industrial production process, 3 different sawn-wood drying techniques were included in the study: air drying (AD), convection kiln drying (KD), and vacuum drying (VD). To achieve sufficiently short processing times, we have combined these available industrial drying techniques. We have tested the possibility of single-phase processing of oak elements to a final moisture content (MC) of 8% before bending ($G_1$, $G_2$) and two-phase processing ($G_3$, $G_4$). In the latter case, phase 1 is performed up to a wood MC of 16%, at which the wood was bent. For high wood MCs, only air drying and convection kiln drying were used, while for drying at low MCs (<16%), vacuum drying was additionally tested. By combining different drying techniques (AD, KD, VD) and the final MC achieved before the bending process (16% and 8%), 4 test groups ($G_1$...$G_4$) were formed (Table 1).

**Table 1.** Classification into process groups ($G_1$...$G_4$) and treatment sequence of test oak samples, with target moisture content, before and/or after solid wood bending.

| Test Group | Processing Procedure | No. of Samples |
|:---:|:---:|:---:|
| **$G_1$** | Air drying (20...25%) → Kiln drying (8%) → **Bending** | 30 |
| $G_2$ | Kiln drying (20...25%) → Vacuum drying (8%) → **Bending** | 30 |
| $G_3$ | Kiln drying (16%) → **HF Bending** → Kiln drying (8%) | 30 |
| $G_4$ | Kiln drying (16%) → **Bending** → Kiln drying (8%) | 30 |

Air drying of $G_1$ test samples took place in the first half of 2022 in the sawmill warehouse at the company's site. The average drying temperature increased from 3 °C in January to 18 °C at the end of May 2022. Subsequently, the $G_1$ samples were conventionally dried in the kiln dryer at normal temperature (<45 °C) and moderate drying gradient (<2.5) until a target value of 8% MC was reached. The solid wood was then steamed and bent.

For the $G_2$ samples, standard convection kiln drying was performed at low temperature from the initial green state to 22% MC. These samples were then placed in a vacuum drying chamber where heat transfer was performed using the hot plate method. Vacuum drying was performed at a temperature as high as 70 °C and an absolute vacuum of nearly 100 mbar to achieve a final value of 8% MC. This was followed by steaming and bending of dried raw oak wood pieces.

The samples in groups $G_3$ and $G_4$ were also kiln dried using the same drying equipment as for samples $G_1$ and $G_2$ (Figure 2a). However, the samples of groups $G_3$ and $G_4$ were dried in 2 stages. First, they were convection kiln-dried as raw elements at low temperature (following the same drying schedule as group $G_2$) until they had a final MC of 16%. The solid wood was then steamed and bent. The bent solid wood elements were then dried in the same kiln dryer at normal temperature to a final MC of 8%, as in $G_1$.

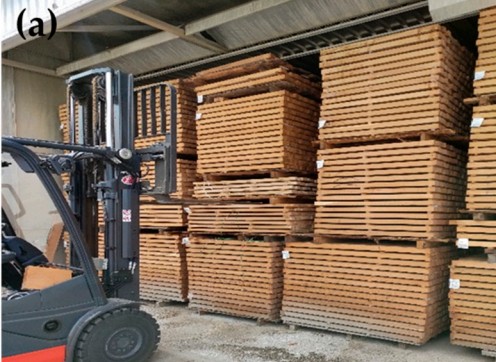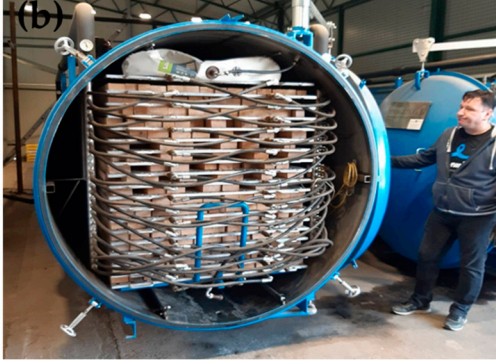

**Figure 2.** Normal temperature convection kiln drying (KD) of test oak samples (**a**) and vacuum drying (VD), using hot plate method for heat transfer (**b**).

#### 2.2.2. Solid Wood Steaming

Before bending, the dried specimens were steamed. The specimens were placed in boilers heated with steam at 130 ± 3 °C and a pressure of 1.8 bar. The process took an average of 3.5 h (±0.5 h). The temperature at the end of steaming, just before the specimens were placed in the bending press, was checked with a non-contact thermal imaging camera (FLIR i60; FLIR Systems AB, Täby, Sweden). The surface temperature of the specimens was above 70 °C, while the temperature inside the specimens was even higher (Figure 3).

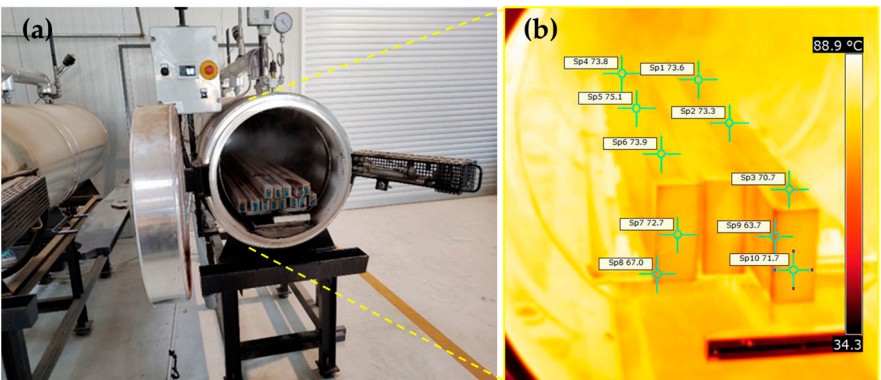

**Figure 3.** Steaming of solid oak specimens in boiler (**a**) and achieved surface temperature (FLIR i60) before the bending (**b**).

#### 2.2.3. Solid Wood Bending

The bending of the oak specimens of groups $G_1$, $G_2$, and $G_4$ was carried out classically in the bending press using a metal belt with end stop on the convex side of the specimens. A mold with a center radius of 150 mm was used for bending. Three specimens each were placed in the press-bending mold. The angle between the two straight parts of the bent specimen was 105°, which corresponded to the desired geometry of the piece of furniture—the armchair (Figure 4). The pressing time was 30 min. Subsequently, the bent specimens were placed in stacks, either for further mechanical processing ($G_1$ and $G_2$) or for additional drying ($G_4$).

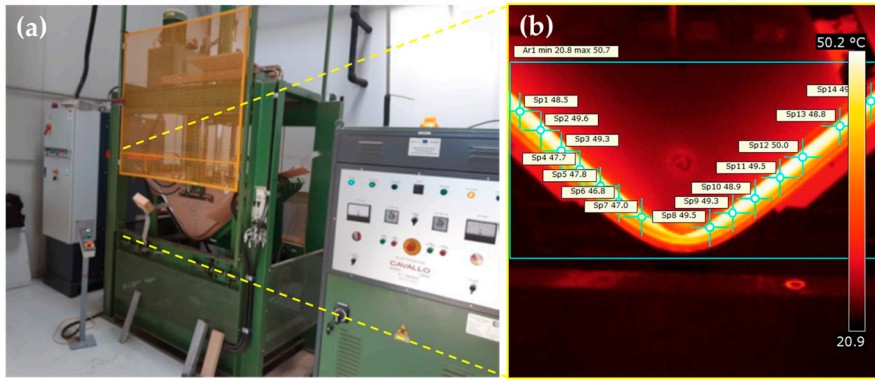

**Figure 4.** Press with bending mold (**a**) and achieved temperature (FLIR i60) in bent oak samples during the bending process of (**b**).

To increase productivity, $G_3$ samples were bent at high frequency (HF). Research on microwave drying of wood has been conducted since the early 1960s and is well-documented [20]. The same press and mold was used for HF bending as for conventional bending. With a 20 kW HF generator, the wood was bent at 10 MHz, heated, and finally dried to 8% MC. The pressing time was on average 30 min.

### 2.3. Quality Assessment of Bent Oak Elements

The quality of the bent oak was determined primarily visually. In the bending area, the tested specimens were carefully inspected in all planes. Bending was confirmed to be successful when the specimens showed no tensile cracks and failures on the convex side and a homogeneous structure on the concave side. Specimens with local bending, buckling, and kneading of the fibers on the concave side of the specimens were also rejected.

The velocity of ultrasound (v) has been added to the quality assessment of bent oak samples with twice-per-piece ($v_S$—straight part, $v_B$—bent region) measurement of the time of flight in the longitudinal direction by Proceq Pundit PL-200PE (Proceq Inc., Scharzenbach, Switzerland) pulse ultrasonic device. The exponential transducers with working 54 kHz frequency were used (Figure 5). The velocity of ultrasound in the straight part of the specimens ($v_S$) and wood density (ρ) were used to determine the modulus of elasticity (MOE) of bent oak specimens (Equation (1)).

$$MOE = ρ \cdot v_S^2 \tag{1}$$

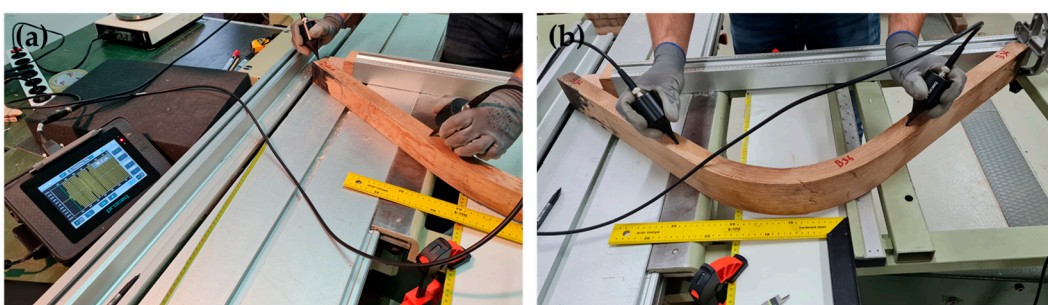

**Figure 5.** Determination of the velocity of ultrasound (Proceq PL-200PE) in the straight part ($v_S$) (**a**) and in the bent region ($v_B$) of solid oak specimens (**b**).

## 3. Results and Discussion

### 3.1. Process Kinetics

Air drying of $G_1$ specimens was the slowest and the longest process, depending on local climatic conditions. Air drying lasted 84 days, during which the wood reached an average MC of 21.9%. We then dried the wood in a convection kiln dryer for an additional 28 days to an average final MC of 8.7%. The drying of group $G_2$ was only slightly faster than that of group $G_1$. In the first part, drying took 56 days to an average MC of 20.8%, followed by 11 days of vacuum drying to a final MC of 8.6%. In both groups, specimen bending was performed at low MC (<10%) (Figure 6).

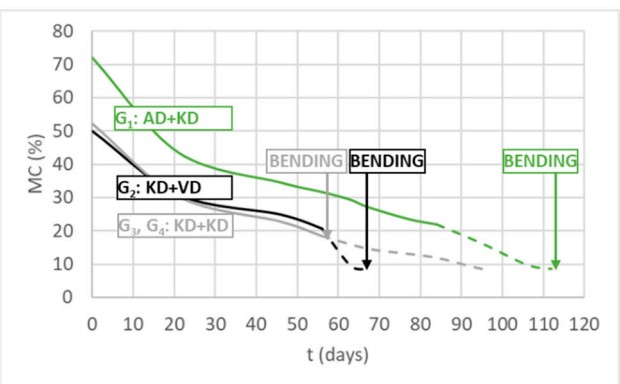

**Figure 6.** Kinetics of drying processes with bending of four tested groups of oak specimens, used for solid wood bending: $G_1$ sequence: 1. air drying (AD) → 2. kiln drying (KD) → 3. bending, $G_2$ sequence: 1. kiln drying (KD) → 2. vacuum drying (VD) → 3. bending, $G_3$, $G_4$ sequence: 1. kiln drying (KD) → 2. bending → 3. kiln drying (KD) (— 1st drying technique, - - - 2nd drying technique).

The drying of the $G_3$ and $G_4$ samples was carried out in the same way as that of the $G_2$ samples in the first phase. An average MC of 16.8% was obtained, and the wood was bent in this condition, using the HF field in group $G_3$ and the conventional method in group $G_4$. The drying of the bent wood of groups $G_3$ and $G_4$ in the second phase in a kiln dryer to a final MC of 8.6% took another 39 days.

### 3.2. Bending Success Rate of Oak Wood

Bending success rates were highest for groups $G_3$ (60.9%) and $G_4$ (42.9%) when specimens were bent at higher MCs (>16%). In these two groups, $G_3$ specimens appear to have lost some success at the expense of HF bending (Figure 7). As other studies have shown, excessive energy input of HF into the bent specimens can lead to an increase in temperature and, due to trapped moisture in the specimens, an increase in vapor pressure that can break down the cellular wood structure [17]. Especially in impermeable wood species such as European oak [21], this can lead to high internal stresses and possibly failure.

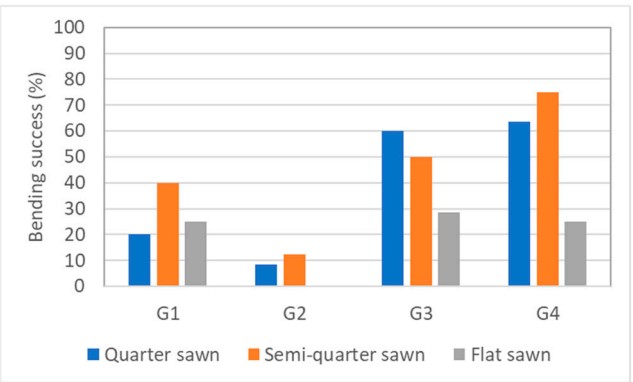

**Figure 7.** Bending success rate of oak wood in each test group ($G_1…G_4$) with respect to orientation of specimens (quartersawn, half-quartersawn, flatsawn).

However, we did not succeed in bending the $G_1$ (28.6%) and $G_2$ (8.3%) samples. It appears that the low moisture content of the bends ($MC_{G1}$ = 8.7%; $MC_{G2}$ = 8.6%) does not allow the necessary plastic deformability that can be achieved when the material exceeds the glass transition temperature [14,15]. At the same time, it is shown that accelerated vacuum drying in stage 2, from 20.8 to 8.6% MC ($G_2$ group), can cause high internal stresses due to the poor permeability of the oak wood tissue [22]. These can lead to additional microstructural defects, which then cause the material to fail during bending. This is reflected in the higher rejection rate of the $G_2$ group compared to $G_1$.

### 3.3. Appearance and Visual Assessment of Defects of Bent Oak Wood

Visual inspection of the rejected bent oak specimens revealed approximately three typical defects, with material failing on the concave side (I), on the convex side (II), or on both sides (III) (Figure 8). On the concave side, local wrinkling, buckling, and kneading of the fibers were observed, most pronounced at the bottom of the curve (Figure 8b, I—failure type). On the opposite, convex side, several of the rejected bends exhibited tensile failure, with both brittle and fiber breaks (Figure 8c, II—failure type). The worst case, where bending was unsuccessful, was represented by the test specimens where local fractures occurred on both the concave and convex sides (Figure 8d, III—failure type).

The various failure modes occurred to about the same extent in all specimen groups examined ($G_1…G_4$). For the individual orientations (quartersawn, quarter- to flatsawn, flatsawn), we could not identify a characteristic failure mode. The main cause of structural failure appears to be poor plasticization of the specimens before or during the bending process. This is also a commonly cited reason for solid wood bending in past research [4,10,18].

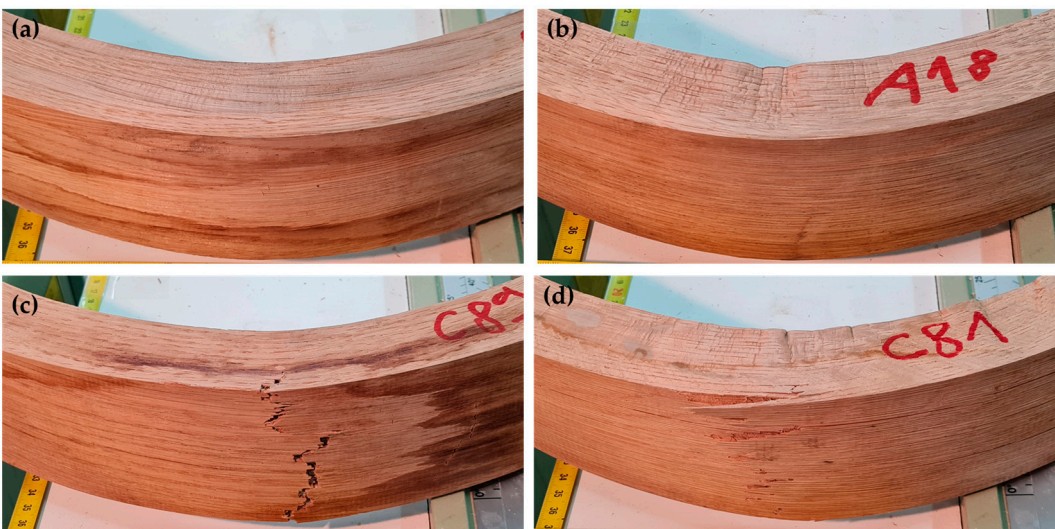

**Figure 8.** Visual assessment of defects on bent oak: (**a**) no defects present, (**b**) wrinkling, buckling, and kneading of fibers on concave side (I—failure type), (**c**) tensile brittle fiber break (II—failure type), (**d**) buckling and kneading of fibers on concave side with additional tensile fiber break on convex side (III—failure type).

### 3.4. Physical and Acoustic Properties of Bent Oak Wood

The average density of oak wood at an average final MC of 8.4% was 634 kg/m$^3$ (CoV = 8.1%; CoV—Coefficient of Variation), which is lower compared to the results of other studies on this species [23–26]. Some studies suggest that the lignin content decreases with increasing density of oak wood, which could affect the bending capacity of the wood [27,28]. No significant differences were found between the mean values of the groups ($\rho_{G1}$ = 640 kg/m$^3$, $\rho_{G2}$ = 635 kg/m$^3$, $\rho_{G3}$ = 624 kg/m$^3$, $\rho_{G4}$ = 638 kg/m$^3$) (Figure 9), which does not directly indicate possible chemical differences between the studied specimens of each group. The latter would have to be confirmed by further investigations.

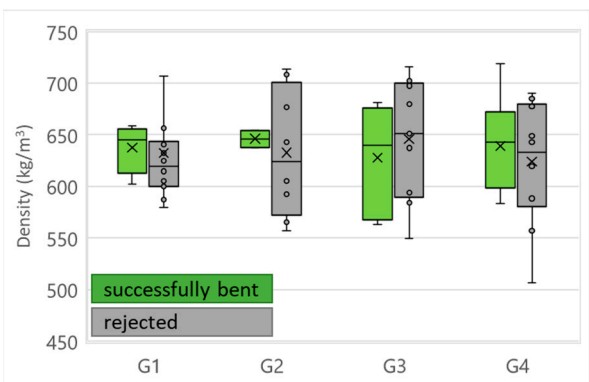

**Figure 9.** Density and its variation of oak wood specimens in tested groups $G_1 \ldots G_4$ ($\times$ mean, $\circ$ outlier).

In the straight part of the successfully bent oak specimens, we determined a mean ultrasonic velocity of less than 4500 m/s ($v_S$) and a corresponding modulus of elasticity (MOE) of less than 13 GPa in all groups studied. There was a downward trend in the mean ultrasound velocity of specimens in groups $G_1$ to $G_4$ ($v_{S-G1}$ = 4470 m/s, $v_{S-G2}$ = 4450 m/s, $v_{S-G3}$ = 4440 m/s, $v_{S-G4}$ = 4220 m/s). We also confirmed a trend of decreasing stiffness from $G_1$ to $G_4$ (MOE$_{G1}$ = 12.8 GPa, MOE$_{G2}$ = 12.8 GPa, MOE$_{G3}$ = 12.5 GPa, MOE$_{G4}$ = 11.4 GPa). However, the differences between the means were not significant for both $v_S$ and MOE, except for $G_4$, where the values were significantly lower than for the others ($G_1$, $G_2$, and

$G_3$; ANOVA, $p < 0.05$) (Figure 10). It seems that part of the best bending performance we have confirmed in group $G_4$ is also due to the slightly lower stiffness of the specimens.

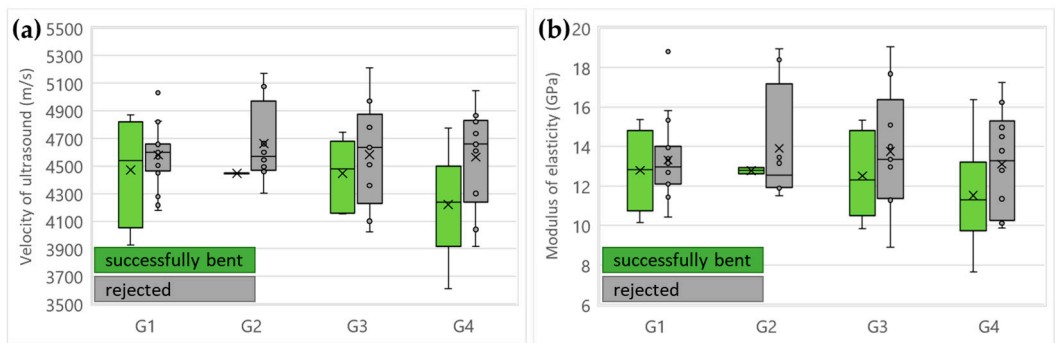

**Figure 10.** Ultrasound velocity ($v_S$) (**a**) and modulus of elasticity (MOE) (**b**) in tested groups $G_1 \ldots G_4$: (**a**) determined in the straight part of bent oak specimens ($\times$ mean, $\circ$ outlier).

In a straight part of the rejected samples, we measured $v_S$ above 4500 m/s and MOE above 13 GPa. We could not find any characteristic trend in the change of values between the groups of rejected specimens ($v_{S\text{-}G1}$ = 4550 m/s, $v_{S\text{-}G2}$ = 4620 m/s, $v_{S\text{-}G3}$ = 4650 m/s, $v_{S\text{-}G4}$ = 4540 m/s; $\text{MOE}_{G1}$ = 13.4 GPa, $\text{MOE}_{G2}$ = 13.7 GPa, $\text{MOE}_{G3}$ = 13.6 GPa, $\text{MOE}_{G4}$ = 13.3 GPa). However, when we compare the values of the rejected specimens with those of the successfully bent specimens, we find that $v_S$ and MOE of these two categories are different in groups $G_3$ and $G_4$. Thus, it can be seen that the initial lower MOE of the specimens has a significant effect on the bending success. Only in groups $G_3$ and $G_4$ was the bending success significantly higher than in $G_1$ and $G_2$, which, in addition to the correspondingly high wood MC, could also be due to the lower MOE of the specimens in $G_3$ and $G_4$. Previous research has also shown that less stiff wood is easier to bend [4,15,18,19].

We measured a much lower average ultrasound velocity in the bent region of the specimens ($v_B$ = 2270 m/s) than in their straight part ($v_S$ = 4505 m/s) (Figure 11a). This is partly due to the curvature itself, as the sound waves can propagate along a shortcut between the two probes and thus across the wood grain where they would otherwise be slower [29–31]. The compression deformation of the wood tissue in the curved region of the specimens could also be a reason. Ultrasound velocities greater than 2300 m/s were measured in the $G_1$ and $G_2$ specimens for both the successfully bent and rejected specimens. No differences in mean values were observed between the two groups ($G_1$ and $G_2$) and the two categories (accepted and rejected). For specimens in groups $G_3$ and $G_4$, the $v_B$ was lower and slightly below 2250 m/s. The differences between the two groups and between the two categories of specimens were also statistically insignificant (*t*-test; $p > 0.05$).

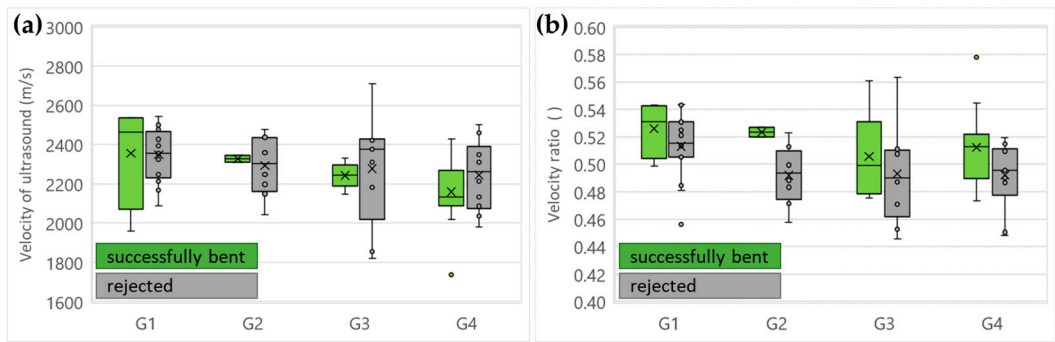

**Figure 11.** Ultrasound velocity in bent region ($v_B$) of oak specimens (**a**) and ratio of ultrasound velocities in straight and bent region ($v_S/v_B$) (**b**) of specimens in tested groups $G_1 \ldots G_4$ ($\times$ mean, $\circ$ outlier).

It was found that the relative ultrasound velocity in the bent region of the specimens ($v_B$), represented as the ratio between the ultrasound velocity in the bent region and in the straight portion ($v_B/v_S$), was related to the bending success of the specimens. This ratio averaged above 0.51 for successfully bent specimens and below 0.50 for rejected specimens, regardless of test group ($G_1...G_4$). Due to considerable variability, the $v_B/v_S$ ratio was significantly lower in the rejected compared to the accepted specimens only in groups $G_2$ and $G_4$. Shear ultrasound waves are commonly used to detect internal defects in some other materials [32], and we intend to include them in future studies.

### 4. Conclusions

Bending of European oak solid wood, by studying the influence of wood moisture content and previous drying methods on bending ability in industrial tests and also using non-destructive techniques, led to the following conclusions:

- In order to achieve low rejection rate in the bending process, the process parameters, i.e., the time and the final moisture content for the different processing stages, must be well-controlled, as the margin between time and moisture content for optimal bending is very narrow.
- A method in which the oak wood is predried in one step to a nominal final moisture content of 8% and the specimens are then bent is not practical because of the low bending deformability and low compressibility on the concave side and tensile ductility on the convex side.
- The study showed that the initial lower MOE in addition to the proper moisture content before bending (MC $\geq$ 16%) significantly affected the bendability and acceptance rate of the oak specimens.
- In addition to visual assessment, the acceptance rate of bent solid oak can be determined non-destructively from the ratio of the ultrasound velocity in the straight and bent region ($v_B/v_S$) of the specimens.

**Author Contributions:** Conceptualization and experiment design, A.S., M.M. and S.P. (Stjepan Pervan); M.M., A.S. and J.Ž. performed the experiments; A.S., M.M. and J.Ž. analyzed the data; validation and formal analysis, J.Ž., M.M. and S.P. (Silvana Prekrat); writing—original draft preparation, A.S. and S.P. (Silvana Prekrat); writing—review and editing, A.S., J.Ž., M.M., S.P. (Stjepan Pervan) and S.P. (Silvana Prekrat); project administration, S.P. (Stjepan Pervan) and S.P. (Silvana Prekrat); funding acquisition, S.P. (Stjepan Pervan), M.M. and S.P. (Silvana Prekrat). All authors have read and agreed to the published version of the manuscript.

**Funding:** The research is part of the project "Development of innovative products from modified Slavonian oak" (KK.01.2.1.02.0031) of Spin Valis d.d. and the partner University of Zagreb, Faculty of Forestry and Wood Technology. The total value of the project is HRK 55,064,343.84, while the amount co-financed by the EU is HRK 23,941,527.32. The project was co-financed by the European Union from the Operational Program Competitiveness and Cohesion 2014–2020, European Fund for Regional Development. This work was also supported by the Ministry of Higher Education, Science and Innovation of the Republic of Slovenia under the program P4-0430 (Forest timber chain and climate change: the transition to a circular bio-economy) and the research project CRP, V4-2016, funded by the Ministry of Agriculture, Forestry and Food of the Republic of Slovenia (MKGP) and the Slovenian Research Agency (ARRS).

**Data Availability Statement:** The data presented in this study are not publicly available due to a non-disclosure agreement.

**Acknowledgments:** Our special thanks to technical assistant Luka Krže for mechanical processing of the samples. We also thank Matija Straže for processing the samples and data collection.

**Conflicts of Interest:** The authors declare no conflict of interest. The funders had no influence on the design of the study, the collection, analysis, or interpretation of the data, the writing of the manuscript, or the decision to publish the results.

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
