# Peer review of "The Influence of Processing Conditions on the Quality of Bent Solid Wood from European Oak"

_forests, doi:10.3390/f14051047_

Round 1
Reviewer 1 Report
Dear authors,
The manuscript provides potentially interesting experimental results of the influence of wood moisture content and stiffness, determined by NDT, drying methods on the bending ability of the oak wood. It was shown that the best quality was obtained with bending specimens bent at a moisture content of at least 16 % and quarter or semi-quartersawn. In addition to visual assessment, the acceptance rate of bent solid oak can be determined non-destructively from the ratio of the ultrasound velocity in the straight and bent region (vB/vS) of the specimens.
Minor points:
1. For possible using suggested approach and comparative analysis, it would be better to add a short explanation of the decrease of stiffness in bent parts for samples of group G4 compared with G1-G3. For straight parts of G4 such difference was not observed.
2. It is well-known that the best bending quality can be obtained for low initial MC. It is not so clear why for G1 and G2 the initial MC for bending was 8%.
3. It would be better to add to Figure 8 the numbers of sample groups (G1-G4).
4. It would be better to add correct scientific names for oak (Quercus robur L., Quercus petraea (Matt.) Liebl.)
Author Response
Reviewers' comments are listed in the attached file in black type; our responses are listed below in red type. We thank the editor and the reviewers for their constructive criticism and valuable comments, which were of great help in revising the manuscript.

Reviewer 2 Report
Dear Authors,
You can find the review in the attached file.
Best regards,
Reviewer

Author Response

(The authors gave the same response as above.)
